



# An 8-day composited 36 km SMAP soil moisture dataset from 1979 to 2015 produced using a random forest and historical CCI data

Haoxuan Yang[1], Qunming Wang[1*], Wei Zhao[2], and Peter M. Atkinson[3,4]

[1]College of Surveying and Geo-Informatics, Tongji University, 1239 Siping Road, Shanghai 200092, China.
[2]Institute of Mountain Hazards and Environment, Chinese Academy of Sciences, Chengdu 610041, China.
[3]Faculty of Science and Technology, Lancaster University, Lancaster LA1 4YR, UK.
[4]Geography and Environment, University of Southampton, Highfield, Southampton SO17 1BJ, UK.

*Correspondence to*: Qunming Wang (wqm11111@126.com)

**Abstract.** Soil moisture (SM) plays a significant role in many natural and anthropogenic systems which are essential to supporting
life on Earth. Thus, accurate measurement and assessment of changes in soil moisture globally is of great value, including long-term historical assessment. Since the on-board cycle and detailed parameters of disparate sensors are different, the European Space Agency established the Climate Change Initiative (CCI) program to harmonize the available multisource SM data, producing long time-series surface SM datasets starting from 1978 to the present. However, the Soil Moisture Active Passive (SMAP) mission, launched in 2015, has shown more satisfactory performance in both spatial accuracy and in capturing pattern of temporal changes. In
this paper, a random forest (RF) model was proposed to extend the superior SMAP dataset historically (named RF_SMAP), using the corresponding CCI data time-series. We assumed that the temporal changes in the SMAP dataset are similar generally to those in the available CCI dataset. Accordingly, the RF model was constructed using the temporal characteristics extracted from the CCI SM v05.2 data (coupled with three terrain characteristics and two location characteristics), which was migrated to the prediction of the RF_SMAP dataset. The available *in-situ* SM data and the real SMAP data from April 2015 to April 2016 were used as references to
validate the predicted RF_SMAP data. It was shown that compared with the CCI dataset, the predicted RF_SMAP dataset is closer to the *in-situ* SM data and the real SMAP data. Moreover, the historical RF_SMAP dataset is more accurate than the widely used Global Land Evaporation Amsterdam Model (GLEAM) dataset in terms of average root mean square error (RMSE), bias (Bias), and Kling-Gutpa efficiency (KGE). Thus, the RF_SMAP dataset was shown to be a reliable substitute for the historical CCI dataset, with an unbiased root mean square error (ubRMSE) of 0.035. The new long time-series RF_SMAP dataset, which will be available to
download, will be of great value for a range of research in applications such as climate assessment, agricultural planning, food insecurity monitoring and drought assessment and monitoring.

## 1. Introduction

Soil moisture (SM) plays a vital role in many fields of earth science. It is a basis of energy exchange between the atmosphere and the land surface (Zhou et al., 2021), and an important consideration in agricultural extensification and intensification to support food
security (Acharya et al., 2019; Rigden et al., 2020). Likewise, the monitoring of climate change (Jaeger and Seneviratne, 2010; Guillod et al., 2015) and drought (Zhou et al., 2017; Fang et al., 2021) require the long time-series SM data as a key input for analysis. SM can also affect the evapotranspiration of vegetation, which further influences the terrestrial carbon cycle (Wu et al., 2020; Humphrey et al., 2021). Consequently, the acquisition of high-quality and long time-series SM data is crucial to various applications. Both ground sensor-measured (Larson et al., 2008) and satellite-derived SM data (Beck et al., 2021) are available freely and
currently cover the globe. These two types of data have disparate characteristics. Sensor-measured SM data are generally considered as the true SM value at the point scale, as the measurement process is fairly direct. Hence, different measurement networks, which





consist of several SM monitoring stations, have been successively installed and used around the world. The International Soil Moisture Network (ISMN) is a key example of *in-situ* data derived from various SM measurement networks for scientific research and applications (Dorigo et al., 2011). However, despite the advantage of direct measurement, the limitation of spatial sparsity is

unavoidable. That is, the measured SM data are provided only at the fixed and limited sensor locations available. In contrast, satellite-derived SM data are spatially continuous in the sense that they provide complete spatial coverage. As a result, satellite-derived SM data have greater application value than sensor-measured SM data, especially across large areas (e.g., at the global or national scale). Over the last few decades, various satellite-derived SM data have been produced with active or passive microwave technology, such as the Advanced Microwave Scanning Radiometer-2 (AMSR2) (Cho et al., 2017; Jin et al., 2018), the

Soil Moisture and Ocean Salinity (SMOS) (Kerr et al., 2001; Piles et al., 2011), the Soil Moisture Active Passive (SMAP) (Entekhabi et al., 2010; Chan et al., 2016), the Climate Change Initiative program of the European Space Agency (CCI) (Dorigo et al., 2015; Gruber et al., 2017; Gruber et al., 2019), the Advanced Scatterometer (ASCAT) (Bartalis et al., 2007; Zhang et al., 2021), and the Advanced Microwave Scanning Radiometer onboard the Earth Observing System (AMSR-E) (Njoku et al., 2003; Feng et al., 2017). These platforms update SM data continuously and provide flexible choices for research in related fields, such as hydrology.

There are obvious differences between the aforementioned satellite-derived SM datasets due to their disparate frequencies of the sensors. The L-band (~1.4 GHz) is considered to be more suitable for monitoring surface SM than other frequencies (e.g., C-band or X-band) (Kerr et al., 2001). Moreover, although both the SMOS and SMAP missions carry an L-band sensor for retrieving SM, the temperature brightness observations of SMOS have a larger radiometric error than those of SMAP (De Lannoy et al., 2015). Thus, the SMAP dataset is a more satisfactory a priori choice than the SMOS dataset (Al-Yaari et al., 2017). In recent studies, it has also

been found that the SMAP dataset (with a spatial resolution of 36 km) is a preferable choice relative to other satellite-derived SM datasets. Ma et al. (2019) evaluated four SM products (i.e., AMSR2, SMAP, SMOS, and CCI) and found that the SMAP product was superior to other SM products in terms of capturing pattern of temporal changes in SM. Kumar et al. (2018) used information theory-based metrics to demonstrate that the error in SMAP retrievals was the minimum amongst the listed SM datasets (i.e., SMAP, AMSR-E, ASCAT, SMOS, and AMSR2). Furthermore, Kim et al. (2018) claimed that compared to ASCAT and AMSR2, SMAP

showed closer relation to the *in-situ* time-series data at the global scale.

In addition to the global assessment, regional assessment, which can describe stability in a specific region and guide further improvement of SM products, also revealed the advantage of SMAP. For example, based on a study in the Huai River Basin, China, Wang et al. (2021) showed that SMAP outperformed SMOS data in both winter (December, January, and February) and summer (June, July, and August). Thus, SMAP can be viewed as one of the optimal SM datasets, currently. However, SMAP is the latest

satellite-derived SM data, which began providing effective data from April 2015 (Chan et al., 2018), and approximately six years of data storage is not sufficient to support long time-series studies. That is, historical SMAP data before April 2015 are not available, and have to be replaced by SM data derived from other sensors. However, differences in physical characteristics are unavoidable for SM data derived from various sensors, including sensor properties (Hosseini and McNairn, 2017; El Hajj et al., 2019; Bergstedt et al., 2020), retrieval principles (Njoku et al., 2002; Piles et al., 2009; Das et al., 2014), and the spatial resolution of the SM data (Peng et

al., 2017; Li et al., 2018; Abowarda et al., 2021).

Compared with the short temporal span of the SMAP dataset, the CCI dataset (with a spatial resolution of 25 km) has the longest temporal span, which contains approximately 40 years of data from November 1978 to the present, although the first CCI SM dataset was publicly released in 2012 (Dorigo et al., 2015; Dorigo et al., 2017). The enormous number of data in the time-series is conducive for accomplishing dynamic monitoring. Ma et al. (2021) monitored the agricultural drought in Southwest China using the CCI

dataset from 1978 to 2016 and found that the duration of drought increased over time. Actually, the CCI dataset was produced by merging SM products collected by various sensors, which synergistically combines the strengths of the individual products (Liu et



al., 2012; Liu et al., 2011). To expand the spatial-temporal coverage and maintain the consistency of data in a long time-series, different versions of the CCI dataset were produced continuously by introducing new SM datasets, optimizing the retrieval algorithm, and improving sensor inter-calibration efforts (Dorigo et al., 2017). As one of the newest versions, the CCI v05.2 has been used

widely. It needs to be emphasized that the CCI v05.2 version firstly includes the SMAP dataset. In addition, there exist two improvements for the CCI v05.2 version compared with the previous versions, including the inter-calibration of AMSR-2 and the retrieval algorithm of radiometer data (Zhao et al., 2021). Although the CCI dataset harmonizes the multiple-sensor datasets to ensure optimal temporal-spatial coverage and the consistency of the data, its accuracy is inevitably affected by the inherent differences between the observed datasets. Therefore, based on the demonstrated advantage of SMAP and CCI, it is of great interest

to restore the historical SMAP data before April 2015 to keep the consistency of the SM characteristics in the temporal domain.

In this paper, we proposed to synthesize a spatially seamless (i.e., 8-day composited) 36 km SMAP dataset at the global scale from January 1979 to March 2015. This was undertaken by transferring the CCI dataset from 1979 to 2015, with a random forest (RF)-based learning model constructed between the CCI dataset before and after April 2015. By assuming that the pattern of temporal changes of the CCI dataset is similar to that of the SMAP data, the trained RF model can be applied to the SMAP data after

April 2015, producing the synthesized SMAP dataset before April 2015. For clarity, the synthesized SMAP dataset is denoted as RF_SMAP in this paper. The predicted RF_SMAP dataset retains the advantage of the observed SMAP dataset. Furthermore, the RF_SMAP dataset is spatially seamless, which helps to address the gap issues in the CCI dataset. The RF_SMAP dataset can support the use of homologous SM data in long-term series studies without the need for multi-sensor SM data, which can help to avoid the uncertainty introduced by differences between sensors. The predicted historical SMAP data from 1979 to 2015 will be released

publicly to support related research based on the need of long time-series SM data at the global scale.

## 2. Data and methods

### 2.1. Data description

#### 2.1.1. SM datasets

The 36 km SMAP and 25 km CCI SM dataset used in this paper can be freely collected from the National Snow and Ice Data Center

(https://nsidc.org/) and the Climate Change Initiative program of the European Space Agency (http://www.esa-soilmoisture-cci.org/), respectively. The SMAP dataset was collected from April 2015 to December 2019, while the CCI dataset was collected from January 1979 to December 2019. Meanwhile, two model-based SM datasets were used as benchmark data for comparison with the predicted historical dataset, including the Global Land Evaporation Amsterdam Model (GLEAM) (Martens et al., 2017) and *SoMo.ml* dataset (Sungmin and Orth, 2021). The GLEAM dataset was generated by satellite

and reanalysis data from 1980 to 2021, and the *SoMo.ml* dataset was derived by the model-based data and ground observation (i.e., *in-situ* data) from 2000 to 2019. In this paper, The GLEAM and *SoMo.ml* datasets were collected from January 1980 to April 2015 and January 2000 to December 2004, respectively. For each type of data, 8-day composited data were considered, which can provide spatially complete SM at the global scale and reduce the interference of the stripes in the daily satellite data. To match the spatial resolution of both datasets, the 25 km datasets were degraded to 36 km. **Table 1** lists the details of used SM products.

**Table 1. The SM datasets used in the study.**

| Product | Spatial resolution | Data version | Sensors type | Data period (mm-dd-yyyy) |
|---------|--------------------|--------------|--------------|--------------------------|
| SMAP | 36 km | Version 6 | Passive | 04-15-2015 to 12-31-2019 |
| CCI | 25 km | v05.2 | Active/passive combined | 01-01-1979 to 12-31-2019 |
| GLEAM | 25 km | Version 3.6a | | 01-01-1980 to 04-14-2015 |



| | | | | |
|---|---|---|---|---|
| *Somo.ml* | 25 km | V1 | | 01-01-2000 to 12-31-2004 |

#### 2.1.2. *In-situ* SM data

*In-situ* data are always used as reference data for the validation of SM products (Colliander et al., 2017; Ford and Quiring, 2019), and they can be acquired freely from ISMN (https://ismn.earth/en/). Since passive radiometers cannot penetrate deeper soil, topsoil moisture (< 0.05 m) retrieved from satellite-derived SM data was used alternatively (Adams et al., 2015; Escorihuela et al., 2010; Raju et al., 1995). That is, all depths of the selected *in-situ* observations were not larger than 0.05 m. For each *in-situ* data point, the 8-day data were averaged to match the temporal resolution of the 8-day composited satellite sensor data. We designed two experiments (denoted as Experiments 1 and 2) using data in different periods to test the effectiveness of the proposed method. Specifically, Experiment 1 used data from 15 April 2015 to 13 April 2016 (denoted as 2015105 to 2016097), and Experiment 2 used data before production of the SMAP dataset (from 1979001 to 2015097). It needs to be stressed that there existed differences in the used *in-situ* data between the two experiments, since the old sensors could be defunct and the new sensors can be installed in other locations in different periods. Figure 1 exhibits the locations of the *in-situ* data in Experiments 1 and 2. Table 2 lists details of the *in-situ* observations.

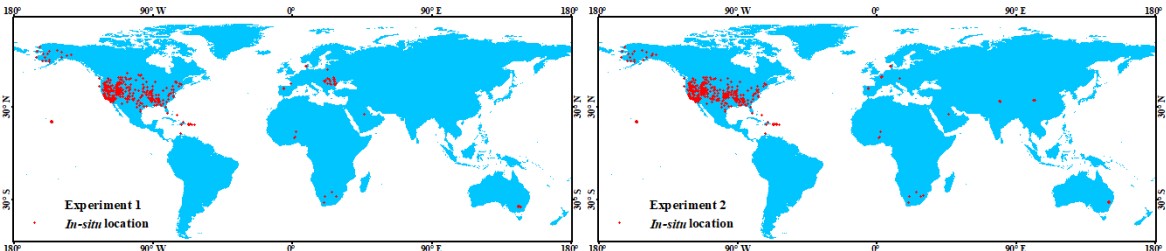

**Figure 1. Locations of the *in-situ* data in Experiments 1 and 2.**

**Table 2. Details of the *in-situ* data used in the experiments.**

| Experiment | Network | Number of *in-situ* points | Location | Period (Format: YYYYDOY) |
|---|---|---|---|---|
| 1 | AMMA-CATCH | 7 | Benin, Niger, Mali | 2015105-2016097 |
| | BIEBRZA_S-1 | 27 | Poland | |
| | FLUXNET-AMERIFLUX | 2 | USA | |
| | FR_Aqui | 3 | France | |
| | HOBE | 29 | Denmark | |
| | OZNET | 19 | Australia | |
| | PBO_H2O | 150 | USA, Saudi Arabia, South Africa | |
| | REMEDHUS | 20 | Spain | |
| | RISMA | 23 | Canada | |
| | RSMN | 20 | Romania | |
| | SCAN | 218 | USA | |
| 2 | AMMA-CATCH | 7 | Benin, Niger, Mali | 2006001-2015097 |
| | ARM | 22 | USA | 2003001-2013313 |
| | HOAL | 33 | Austria | 2013201-2015097 |
| | HOBE | 32 | Denmark | 2011065-2015097 |
| | MAQU | 27 | China | 2008169-2014305 |
| | NGARI | 23 | China | 2011209-2014313 |
| | ORACLE | 3 | France | 1997193-2013241 |
| | PBO_H2O | 150 | USA, Saudi Arabia, South Africa | 2012001-2015097 |
| | REMEDHUS | 24 | Spain | 2005081-2015097 |
| | SASMAS | 14 | Australia | 2006001-2007361 |
| | SCAN | 226 | USA | 1996145-2015097 |



### 2.1.3. Terrain data

The terrain is a crucial factor to affect the variety of soil properties, and plays a significant role in many edaphic studies. We used the
digital elevation models (DEM) to characterize terrain information. The DEM data used in this paper were processed using Google Earth.

### 2.2. Method

### 2.2.1. RF model

The RF model is a multiple decision tree-based ensemble method, which can characterize the relation between independent and
dependent variables reliably with nonlinear regression (Breiman, 2001; Grimm et al., 2008). For the RF model, a bootstrap-based sampling method was used to select the training samples of each tree (approximately two-thirds of all the inputs) for each tree of the model. The remaining one-third of the inputs did not participate in the training process and acted as out-of-bag (OOB) data to validate the constructed model for each bootstrap-based sampling process (Hu et al., 2020; Meng et al., 2020). In addition, the number of regression trees in the forest (n_tree) was also a vital parameter. Guided by the OOB error, n_tree in this research was set
to 200 (Zhao et al., 2018).

### 2.2.2. Characteristics extraction

Surface SM is affected directly by precipitation, and intra-annual variation in precipitation is closely related to seasonal changes (e.g., for the Southern Hemisphere, the precipitation in winter is less than that in summer). Consequently, it can be assumed that temporal variation in surface SM is associated with seasonal changes. Seasonal change is a periodic process across an entire year that is
expected to repeat in subsequent years. Accordingly, the surface SM shows approximate periodic inter-annual variation, exclusive of the occurrence of abnormal climate changes. Therefore, the data selected from an entire year are assumed to have a complete characterization of the temporal pattern. It is noted that, however, some SM data in an entire year include unavoidable spatial gaps, bringing the difficulty in model construction. Moreover, abundant precipitation can easily affect surface SM in some areas, resulting in the abnormal change in the temporal pattern. These two points suggest that the original SM data are unsuitable for direct training.
Thus, a highly comparative time-series analysis (hctsa)-based method (an advanced framework for digging time-series information from data distribution, correlation, and information theory) was adopted to extract spatial seamless characteristics of SM (Fulcher et al., 2013; Fulcher and Jones, 2017). Based on this method, the stable change of the temporal pattern can be preserved as much as possible. For each year, 17 hctsa-based characteristics (HCs) were extracted from original SM time-series. To fully utilize the original CCI and SMAP data, the HCs extracted from different years were jointly used in the input of the RF model. Meanwhile,
three terrain characteristics (TCs) and two location characteristics (LCs) were also considered. The detailed description of these characteristics is presented in **Table 3**.

Table 3. Extracted characteristics for constructing the input of the RF model.

| Characteristic types | Detailed characteristics |
|---|---|
| Hctsa-based characteristics (HCs) | Maximum of the time-series data<br>Minimum of the time-series data<br>Norm mean of the time-series data<br>Median of the time-series data<br>Geometric mean of the time-series data |



| | Harmonic mean of the time-series data |
|---|---|
| | Root mean square of the time-series data |
| | Midhinge of the time-series data |
| | Mean of the trimmed time-series (the 5% of highest and lowest values are trimmed) |
| | Mean of the trimmed time-series (the 10% of highest and lowest values are trimmed) |
| | Mean of the trimmed time-series (the 20% of highest and lowest values are trimmed) |
| | Standard deviation of the time-series data |
| | Interquartile range of the time-series data |
| | Mean absolute deviation of the time-series data |
| | Median absolute deviation of the time-series data |
| | Coefficient of variation of the time-series data |
| | Pearson skewness of the time-series data |
| Terrain characteristics (TCs) | Aspect |
| | Elevation |
| | Slope |
| Location characteristics (LCs) | Latitude |
| | Longitude |

### 2.2.3. Reconstruction of historical RF_SMAP dataset

Two experiments were performed in this research, which focused on the predictions of the RF_SMAP over the period with true SMAP available (April 2015 to April 2016, or 2015105 to 2016097) and the historical period without the SMAP dataset (January 1979 to April 2015, or 1979001 to 2015097).

The full CCI SM dataset is available from January 1979 to December 2019. The relation between the CCI data and the characteristics extracted in Section 2.2.2 can be characterized by a learning model based on RF. Specifically, it is expressed explicitly as follows:

$$\mathbf{CCI}_t = f(\mathbf{HCs_c}, \mathbf{TCs}, \mathbf{LCs}) \tag{1}$$

where $\mathbf{CCI}_t$ is the known CCI SM data (output of training data, that is, label) at a time before April 2016 (Experiment 1: from April 2015 to April 2016; Experiment 2: from January 1979 to April 2015), and $\mathbf{HCs_c}$ are the temporal characteristics extracted based on the known CCI SM time-series (Experiment 1: from April 2016 to December 2019; Experiment 2: from April 2015 to December 2019). $\mathbf{TCs}$ and $\mathbf{LCs}$ are the terrain and location characteristics illustrated in Section 2.2.2.

Although there are inevitable differences between the SMAP and CCI SM data, for each pixel, the variation in SM for the two datasets is similar. We selected randomly two lines (based on the norm mean of the time-series data for the HCs in 2019) and six pixels (based on the SM time-series) at the global scale to exhibit the pattern of changes in the two SM datasets in terms of temporal and spatial domains. As shown in Figure 2, the values of the two SM datasets are different, but they are similar in general pattern of changes. Based on this similar pattern of changes, we assume that the pattern of changes in CCI SM data can be transferred to SMAP data. Therefore, for a time before April 2015, the prediction of the SMAP dataset (denoted as RF_SMAP dataset) can be viewed as a function (i.e., the operator $f$ characterizing the nonlinear relationship in Eq. (1)) of the inputs of SMAP data. It is notable that the input data for the prediction model needs to be acquired in the same period as that for the training model. Thus, the RF_SMAP at a time from April 2015 to April 2016 (Experiment 1) and January 1979 to April 2015 (Experiment 2) can be predicted based on Eq. (2):

$$\widehat{\mathbf{SMAP}}_t = f(\mathbf{HCs_s}, \mathbf{TCs}, \mathbf{LCs}) \tag{2}$$

where $\mathbf{HCs_s}$ are the input data extracted from SMAP time-series (the acquired period of $\mathbf{HCs_s}$ is the same as that of $\mathbf{HCs_c}$), and $\widehat{\mathbf{SMAP}}_t$ is the prediction of SMAP (i.e., the predicted RF_SMAP data). Function $f$ is fitted from the learning model in Eq. (1). The prediction process of the RF_SMAP dataset is shown in Figure 3. For a prediction time t, the specific steps are listed as follows:



(1)   To match the spatial resolution of 36 km of the SMAP data, the CCI data were upscaled from 25 km to 36 km by the nearest neighbor method.

        (2)   The characteristics extracted from CCI time-series (i.e., **HCs$_c$**), **TCs**, and **LCs** were combined as the input training data, while the known CCI data at the corresponding prediction time $t$ were used as the output (i.e., label).

        (3)   The selected input and output of training data were used to train the RF model.

(4)   The corresponding characteristics extracted from SMAP time-series (i.e., **HCs$_s$**), **TCs**, and **LCs** were used as the input of the trained RF model. Then, the RF_SMAP dataset at time $t$ can be predicted.

        (5)   The above steps were repeated for each time in the period from January 1979 to April 2015. Then, the RF_SMAP dataset, as a long time-series, can be acquired.

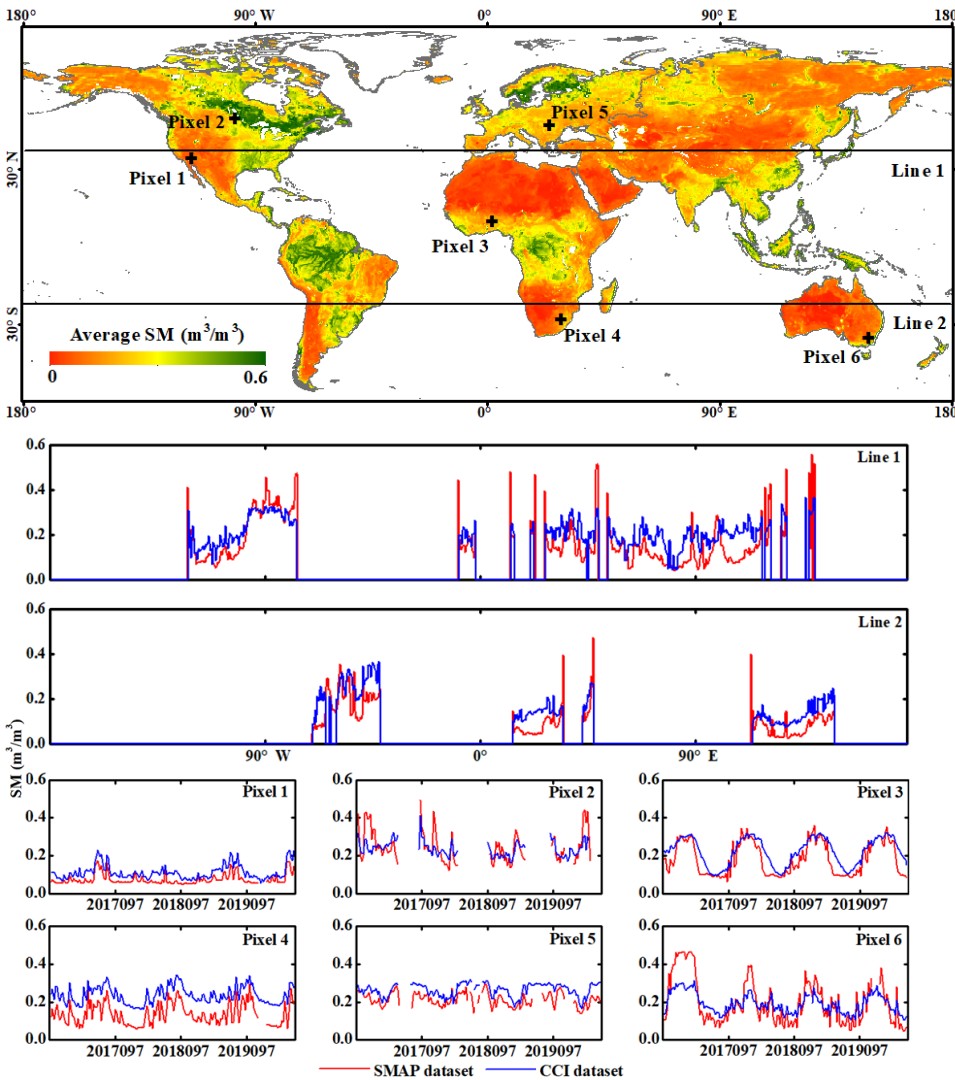

**Figure 2. The patterns of changes in SMAP and CCI SM in temporal and spatial domains.**






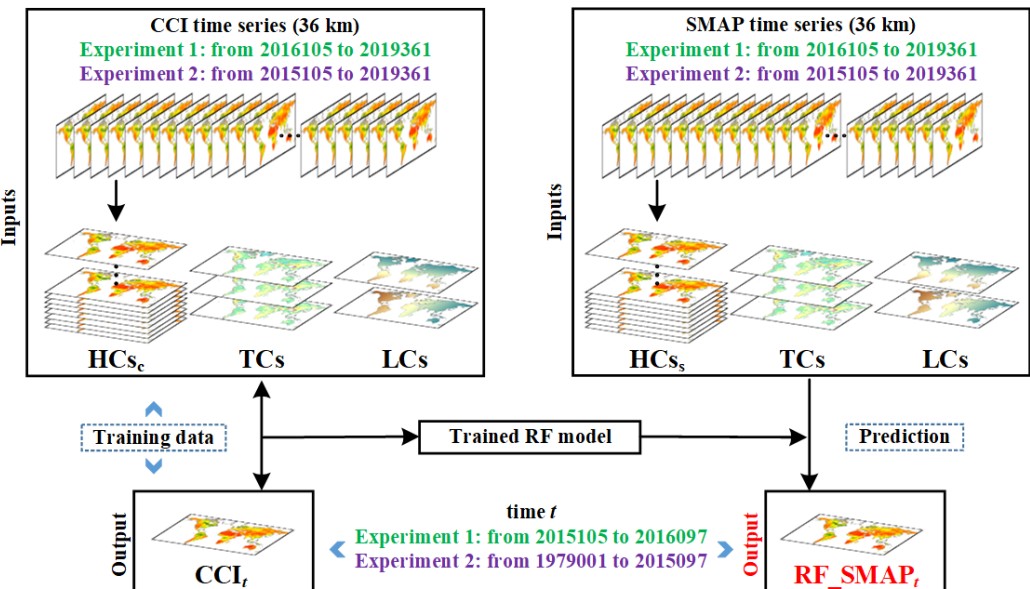

**Figure 3. The prediction process of the RF_SMAP dataset on a date.**

### 2.3. Validation method

For Experiment 1, where the SMAP dataset from 2015105 to 2016097 was predicted, the real SMAP dataset is known perfectly. Thus, the SMAP dataset was used for validation of the prediction directly. For both Experiments 1 and 2, the *in-situ* data were also used to validate the predicted RF_SMAP data. The validation was performed separately for each network listed in Table 2. Specifically, each daily *in-situ* data point in each network was averaged to match the 8-day composited SM of SMAP and CCI. Furthermore, the 8-day composited *in-situ* data in each network were averaged to present the data at the network level. Four

statistical metrics were used for quantitative evaluation, including the correlation coefficient (CC), root mean square error (RMSE), bias (Bias), and unbiased root mean square error (ubRMSE). In addition, two widely used indicators were applied to evaluate the time-series, that is, the Kling-Gutpa efficiency (KGE) (Gupta et al., 2009) and Spearman rank correlation (SRC) (Ahmed Ii and Pradhan, 2019). The KGE can comprehensively consider correlation, variability bias, and mean value bias between simulated and observed time-series. The SRC can measure the strength and direction of monotonic association between two different time-series.

### 3. Experiments and results


### 3.1. Experiment 1

   We predicted 46 scenes of 8-day composited SMAP data from 2015105 to 2016097. The true SMAP dataset, CCI dataset and the predicted RF_SMAP dataset of four days were selected randomly to exhibit in Figure 4. Three main points can be observed. First, there are noticeable differences between the CCI and true SMAP images. Generally, the CCI SM values range across a smaller

interval than that for SMAP. More precisely, for pixels with values very close to the largest value of 0.6 in the SMAP dataset, the corresponding values in the CCI dataset are obviously smaller than 0.6. For pixels with the smallest values (i.e., those close to 0) in the SMAP dataset, the CCI SM values are obviously larger. This conditional bias is mainly attributed to the harmonization process in producing the CCI dataset, which minimizes the difference between the data of various sensors by tuning their values. Second, compared to the CCI dataset, the predicted RF_SMAP images are much closer to the true SMAP images. The advantage lies in





reconstruction of both spatial texture and individual SM values. Specifically, the color of the RF_SMAP images is obviously closer
to the SMAP image, and the difference in the texture of some regions is much smaller, such as the orange parts in North America and
Asia. Third, the RF_SMAP dataset fills some gaps observed in the CCI dataset. This is because the spatial seamless $\mathbf{HCs_s}$ were used
as the input of the RF-based prediction model, producing spatial seamless RF_SMAP dataset accordingly.

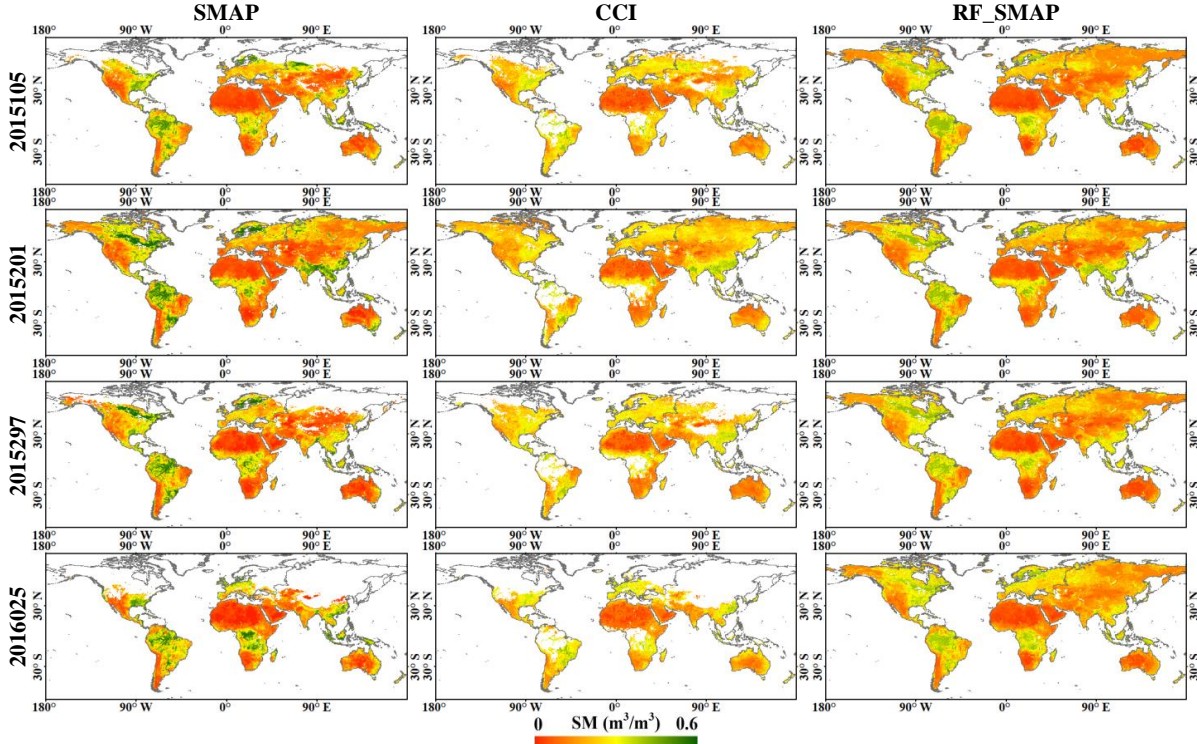

**Figure 4. The three satellite-derived SM datasets in Experiment 1.**

As shown in Figure 5, the true SMAP dataset was used as the reference to evaluate the CCI dataset and RF_SMAP dataset based on
the four accuracy metrics. Note that for fair comparison, the common effective part (i.e., without gaps) of the three datasets was
considered. It can be seen clearly that the accuracy of RF_SMAP prediction is consistently greater than for CCI on each day. The
statistical metrics were averaged from 2015105 to 2016097, and the results are shown in Table 4. The predicted RF_SMAP dataset
has an average CC of 0.926, which is 0.194 larger than that for the CCI dataset (with an average CC of 0.732). Both average RMSE
and ubRMSE of the RF_SMAP dataset are approximately 0.040 smaller than that of the CCI dataset. The average Bias of RF_SMAP
dataset is 0.007, which is also much closer to the reference than that of the CCI dataset (with an average Bias of -0.014).

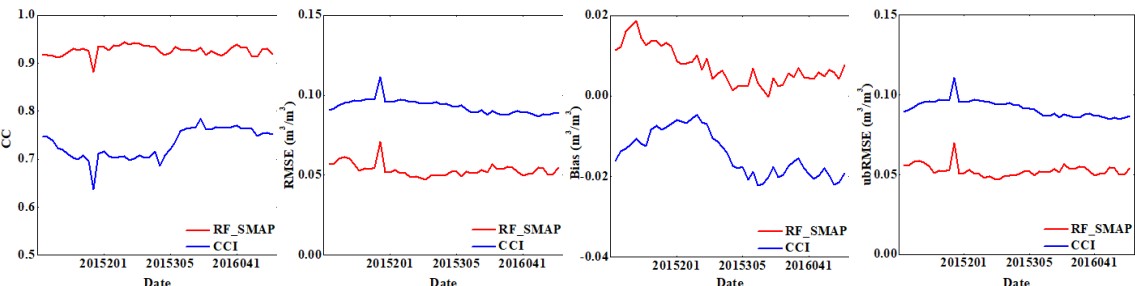

**Figure 5. Statistical metrics for accuracy assessment of the CCI and RF_SMAP datasets against the SMAP dataset in Experiment 1.**





**Table 4. Averaged accuracy indices for all days in Figure 6.**

|  | CC | RMSE | Bias | ubRMSE |
|---|---|---|---|---|
| RF_SMAP | 0.926 | 0.053 | 0.007 | 0.053 |
| CCI | 0.732 | 0.093 | -0.014 | 0.092 |


The *in-situ* data at the 11 networks (see Table 2) were also used to evaluate the observed CCI and the predicted RF_SMAP dataset, as shown in Figure 6. There are some null values in the 8-day composited *in-situ* data, as some of them were not available in the period (not observed by the sensor or acquired with limited quality). It needs to be emphasized that the *in-situ* SM values at the BIEBRZA_S-1 work are larger than those of the corresponding CCI and RF_SMAP datasets and also *in-situ* SM values at other

networks. This is because the BIEBRZA_S-1 network is located at a wetland (including grassland and marshland), and the occurrence of floods is common (Dabrowska-Zielinska et al., 2018). For the networks located at high latitude (e.g., HOBE and RISMA), it is observed that the predicted RF_SMAP dataset can describe the pattern of temporal changes more accurately than the CCI dataset. This is because the CCI dataset at high latitude includes a certain of spatial gaps, resulting in the unstable composite SM data.

Overall, the CCI and RF_SMAP datasets are able to describe the pattern of temporal changes of SM at different locations. However, there are noticeable differences between the *in-situ* data and the satellite-derived observations, revealing the inherent uncertainty in satellite-derived observations. Generally, several types of vegetation with different characteristics cover the topsoil, which influence directly the reliability of the SM retrieved from satellite sensor data. Conversely, the *in-situ* data were measured directly from the topsoil, which avoids surface interference.

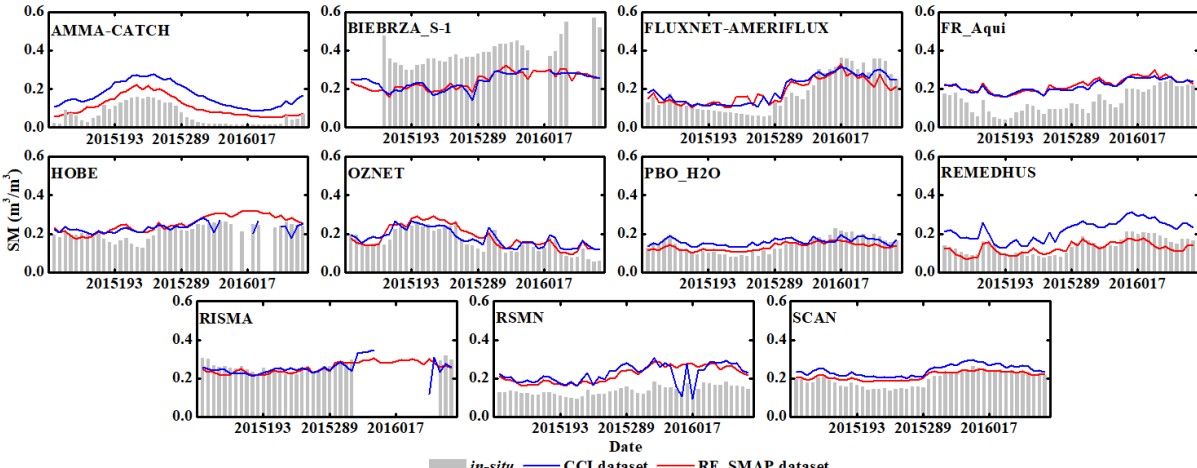


**Figure 6. The *in-situ*, CCI, and RF_SMAP SM datasets at the nine networks in Experiment 1.**

Following Figure 6, the relation in terms of the scatterplots between the CCI or RF_SMAP and the *in-situ* SM datasets is shown in Figure 7. To evaluate directly the accuracies, Table 5 summarizes Figure 7, which lists the four statistical metrics of the CCI and predicted RF_SMAP dataset. The average KGE of the RF_SMAP dataset is 0.452, which is 0.164 larger than that of the CCI dataset.

The predicted RF_SMAP dataset has average CC and SGC of 0.793 and 0.828, which is 0.006 and 0.003 smaller than those of the CCI dataset. The average RMSE of the RF_SMAP dataset is 0.059, which is 0.011 smaller than that of CCI. RF_SMAP and CCI have an average Bias of -0.009 and -0.028, respectively. In addition, the ubRMSE of the RF_SMAP dataset is 0.032, which is 0.001 smaller than that of the CCI dataset. Thus, it can be concluded that RF_SMAP is closer to the *in-situ* data than CCI.




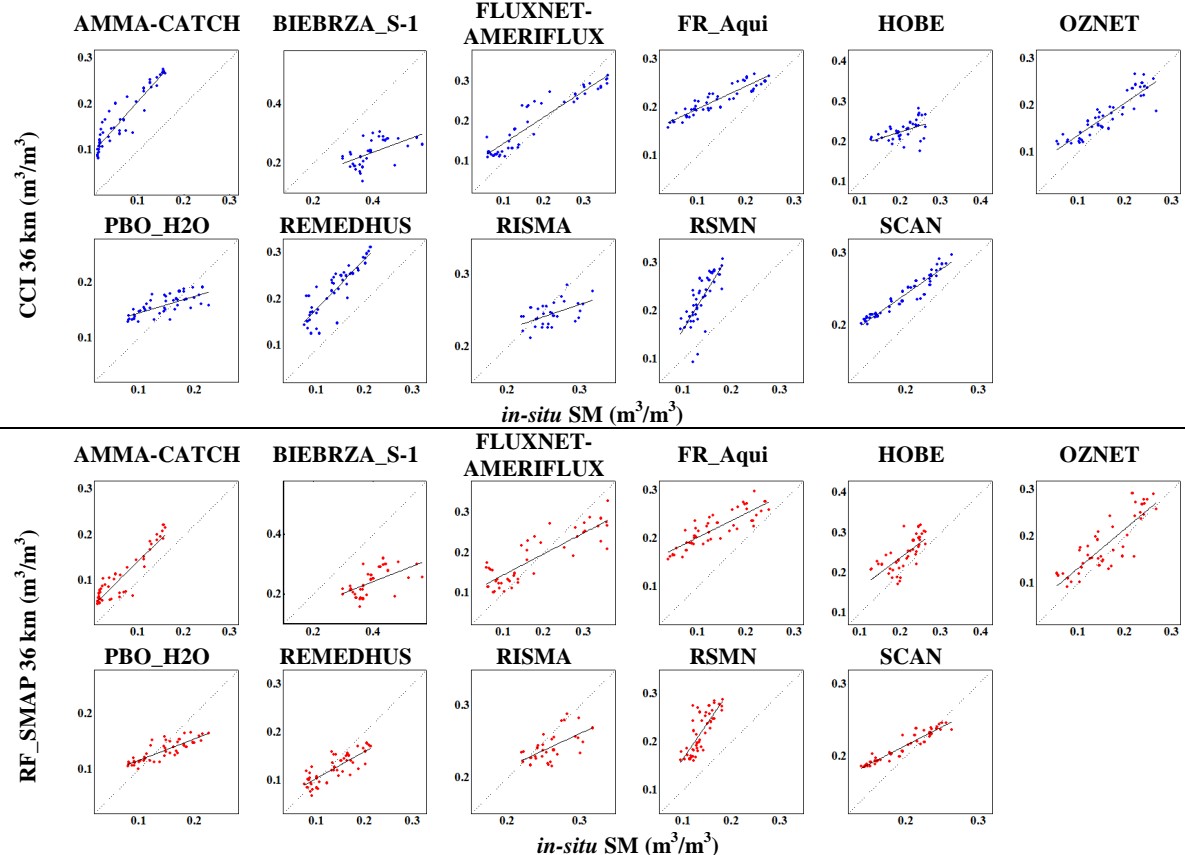

**Figure 7. The scatterplots between the *in-situ* data and SM datasets in Experiment 1.**

**Table 5. Statistical metrics for accuracy assessment of the data at each network in Experiment 1 (using the *in-situ* data as reference).**

|  | CC | | RMSE ($m^3/m^3$) | | Bias ($m^3/m^3$) | |
|---|---|---|---|---|---|---|
| Networks | CCI | RF_SMAP | CCI | RF_SMAP | CCI | RF_SMAP |
| AMMA-CATCH | **0.934** | 0.926 | 0.105 | **0.049** | -0.103 | **-0.044** |
| BIEBRZA_S-1 | 0.561 | **0.582** | 0.170 | **0.164** | 0.161 | **0.154** |
| FLUXNET-AMERIFLUX | **0.936** | 0.860 | **0.047** | 0.059 | -0.017 | **-0.006** |
| FR_Aqui | **0.917** | 0.843 | **0.085** | 0.091 | **-0.078** | -0.084 |
| HOBE | 0.500 | **0.676** | **0.039** | 0.047 | **-0.019** | -0.035 |
| OZNET | **0.903** | 0.847 | **0.032** | 0.039 | **-0.018** | -0.022 |
| PBO_H2O | **0.911** | 0.842 | 0.042 | **0.029** | -0.036 | **0.010** |
| REMEDHUS | **0.871** | 0.803 | 0.086 | **0.028** | -0.082 | **0.012** |
| RISMA | 0.535 | **0.589** | **0.027** | 0.028 | **0.015** | 0.018 |
| RSMN | 0.764 | **0.793** | 0.093 | **0.088** | -0.086 | **-0.083** |
| SCAN | **0.963** | 0.957 | 0.046 | **0.025** | -0.044 | **-0.016** |
| Average | **0.799** | 0.793 | 0.070 | **0.059** | -0.028 | **-0.009** |

| | ubRMSE ($m^3/m^3$) | | KGE | | SGC | |
|---|---|---|---|---|---|---|
| Networks | CCI | RF_SMAP | CCI | RF_SMAP | CCI | RF_SMAP |
| AMMA-CATCH | 0.033 | **0.020** | -0.604 | **0.313** | **0.972** | 0.949 |
| BIEBRZA_S-1 | 0.056 | **0.055** | 0.312 | **0.341** | 0.624 | **0.642** |





| | | | | | | |
|---|---|---|---|---|---|---|
| FLUXNET-AMERIFLUX | **0.049** | 0.054 | **0.679** | 0.585 | **0.938** | 0.829 |
| FR_Aqui | **0.034** | 0.035 | 0.244 | **0.246** | **0.949** | 0.906 |
| HOBE | 0.033 | **0.032** | 0.371 | **0.630** | 0.491 | **0.744** |
| OZNET | **0.027** | 0.033 | 0.723 | **0.794** | **0.904** | 0.813 |
| PBO_H2O | 0.031 | **0.028** | 0.348 | **0.433** | 0.744 | **0.847** |
| REMEDHUS | **0.025** | 0.025 | 0.349 | **0.627** | **0.860** | 0.771 |
| RISMA | 0.022 | **0.021** | 0.407 | **0.533** | 0.860 | **0.863** |
| RSMN | 0.035 | **0.028** | -0.312 | **-0.063** | **0.823** | 0.799 |
| SCAN | **0.014** | 0.019 | **0.649** | 0.530 | **0.972** | 0.949 |
| Average | 0.033 | **0.032** | 0.288 | **0.452** | **0.831** | 0.828 |


### 3.2. Experiment 2

In Experiment 2, the historical SMAP dataset from 1979001 to 2015097 was recovered by the proposed RF model. We predicted

1661 scenes of the RF_SMAP dataset, and eight scenes of data could not be predicted due to the absence of the CCI dataset (i.e., the

dataset on 1981273, 1983273, 1984225, 1986089, 1986097, 1987345, 1987353, and 1988001). Meanwhile, as a benchmark dataset,

the GLEAM dataset was used in Experiment 2. As shown in Figure 8, CCI, GLEAM, and RF_SMAP datasets on five days were

selected randomly for visual comparison. It should be noted that although the 8-day composited CCI dataset was used in this

research, many stripe gaps are observed from the earlier CCI dataset (before September 1987). This is because the stripe gaps were

produced by the short swath width of the Nimbus7 SMMR radiometer (i.e., 780 km) (Owe et al., 2008). Additionally, the operation

of a single radiometer without the aid of other ones is another reason. After September 1987, numerous stripe gaps were gradually

filled with the longer swath width and the appearance of more sensors (Dorigo et al., 2017; Dorigo et al., 2015). Actually, the

predicted RF_SMAP dataset accomplishes a more complete spatial coverage based on the spatial seamless characteristics. It is seen

that the GLEAM dataset also can provide complete spatial coverage. However, the GLEAM data on 1979001 cannot be acquired,

because the GLEAM dataset was started in 1980.

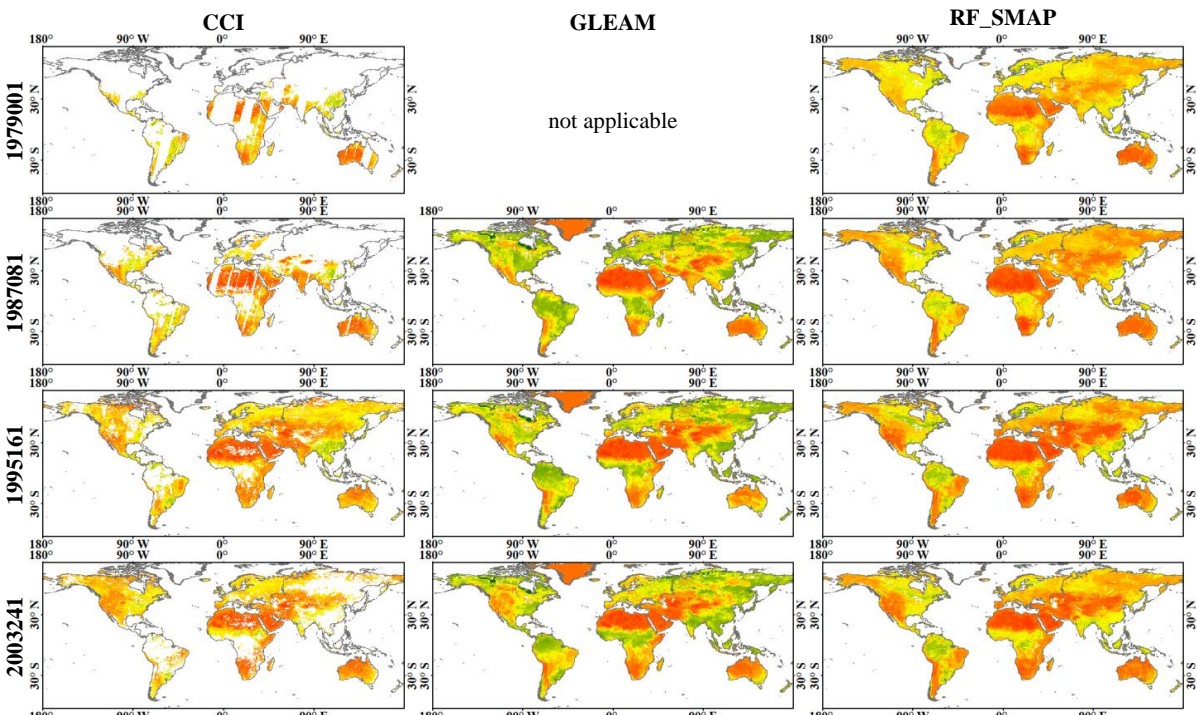

Earth System
Science
Data

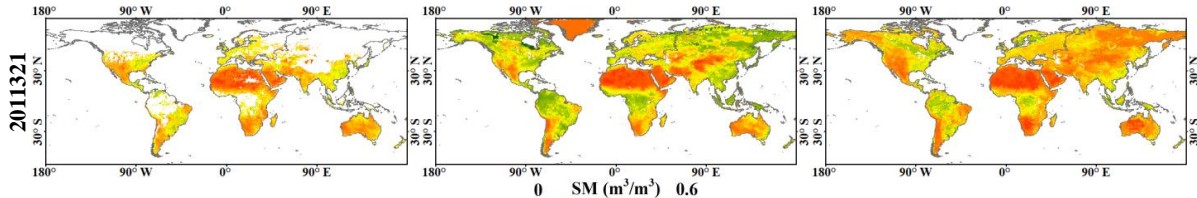

**Figure 8. The original CCI, GLEAM, and predicted RF_SMAP SM data in Experiment 2.**

To compare the accuracy of the CCI, GLEAM, and predicted RF_SMAP datasets, the *in-situ* data at 11 networks were used for
accuracy assessment, as shown in Figure 9. All three datasets can describe clearly the historical periodical changes and pattern of
temporal changes in SM. It needs to be highlighted that the displayed period is different for each network, as these networks have
different on-board periods for acquiring data. Thus, the exhibited periods for the satellite-derived SM data need to match the
on-board periods of these networks. In addition, the spatial gaps of the CCI data cause interruptions in the CCI time-series for

MAQU, NGARI, and HOAL. In contrast, the RF_SMAP and GLEAM dataset generally has a more continuous profile.

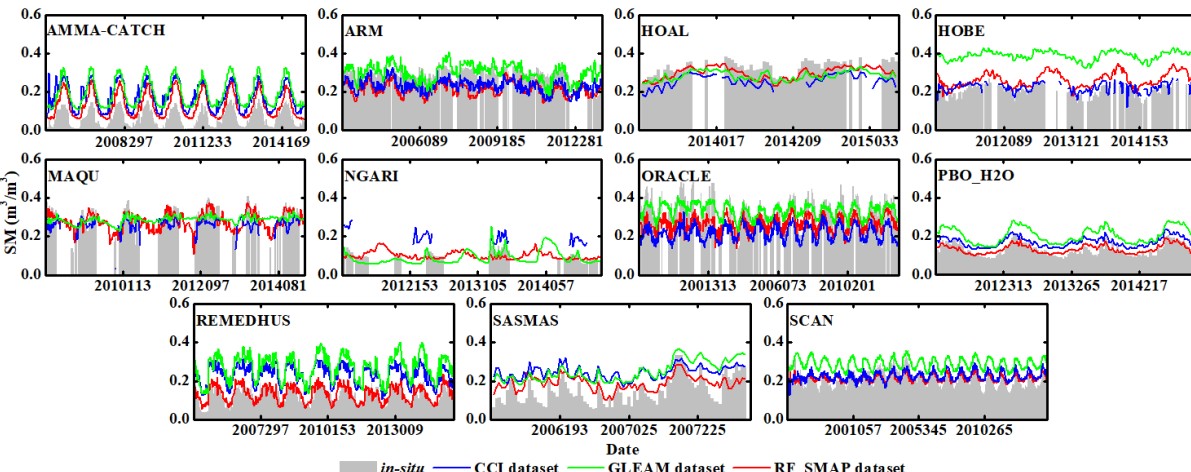

**Figure 9. The *in-situ*, CCI, GLEAM, and RF_SMAP SM datasets at the 11 networks in Experiment 2.**

The scatterplots between the *in-situ* data and the three SM datasets are exhibited in Figure 10. The corresponding statistical
accuracies are listed in Table 6. We found that the RF_SMAP dataset is closer to the *in-situ* data than that of the CCI dataset based on

the statistical metrics. Specifically, the average KGE of the RF_SMAP dataset is 0.414, which is 0.159 and 0.249 larger than that of
the CCI and GLEAM datasets, respectively. The average RMSE and ubRMSE of the RF_SMAP dataset are 0.037 and 0.035, which
are 0.039 and 0.002 smaller than those of the CCI dataset. Moreover, the average Bias of RF_SMAP is 0.006, which is more
satisfactory than that of the CCI and GLEAM datasets. It is seen that, however, the GLEAM dataset has the largest CC and SRC
among the three datasets.

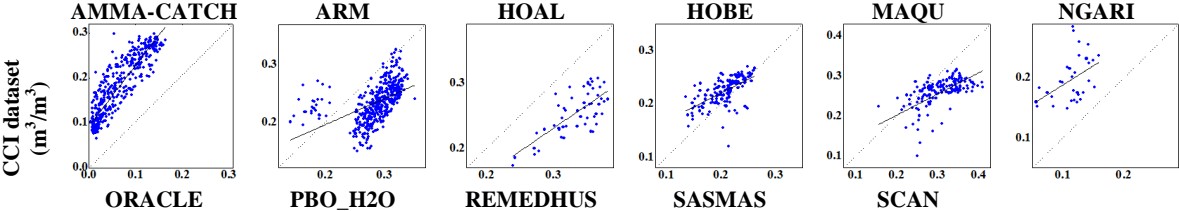



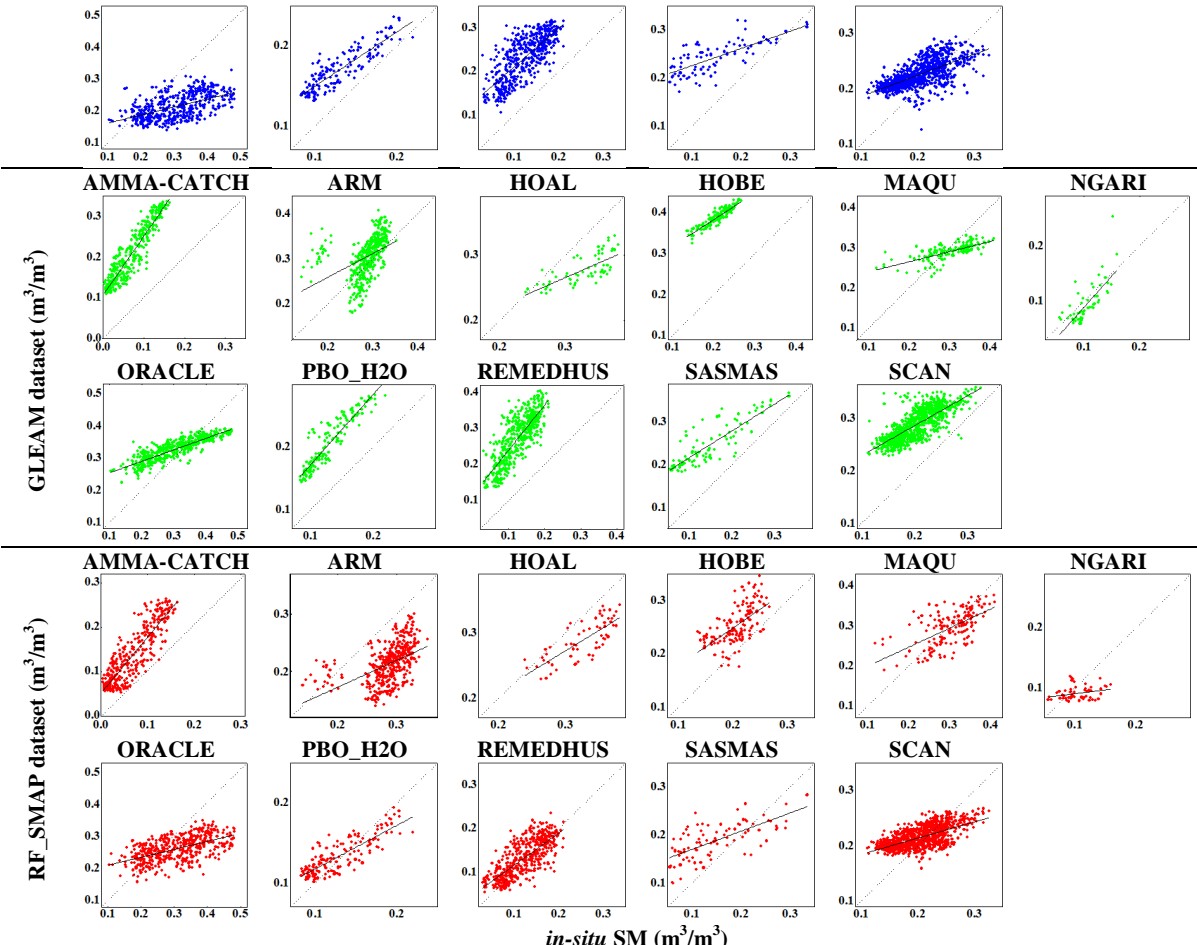

**Figure 10. The scatterplots between the *in-situ* data and SM datasets in Experiment 2.**

**Table 6. Statistical metrics for accuracy assessment of the data at each network in Experiment 2 (using the *in-situ* data as reference).**

| Networks | CC | | | RMSE (m³/m³) | | | Bias (m³/m³) | | |
|---|---|---|---|---|---|---|---|---|---|
| | CCI | GLEAM | RF_SMAP | CCI | GLEAM | RF_SMAP | CCI | GLEAM | RF_SMAP |
| AMMA-CATCH | 0.895 | **0.945** | 0.904 | 0.120 | 0.136 | **0.034** | -0.114 | -0.133 | **-0.068** |
| ARM | **0.491** | 0.433 | 0.490 | 0.063 | 0.046 | **0.034** | 0.052 | **-0.018** | 0.073 |
| HOAL | 0.725 | 0.729 | **0.741** | 0.083 | 0.059 | **0.025** | 0.079 | 0.054 | **0.041** |
| HOBE | 0.609 | **0.918** | 0.621 | 0.030 | 0.180 | **0.025** | **-0.017** | -0.180 | -0.047 |
| MAQU | 0.606 | **0.684** | 0.597 | 0.064 | 0.041 | **0.041** | 0.049 | 0.016 | **0.010** |
| NGARI | 0.450 | **0.724** | 0.436 | 0.090 | **0.026** | 0.035 | -0.083 | **0.003** | 0.025 |
| ORACLE | 0.564 | **0.870** | 0.550 | 0.084 | **0.061** | 0.071 | 0.046 | 0.020 | 0.092 |
| PBO_H2O | 0.896 | **0.924** | 0.885 | 0.042 | 0.077 | **0.025** | -0.040 | -0.076 | **-0.005** |
| REMEDHUS | 0.751 | **0.808** | 0.741 | 0.113 | 0.151 | **0.034** | -0.108 | -0.146 | **-0.009** |
| SASMAS | 0.755 | **0.871** | 0.712 | 0.105 | 0.104 | **0.052** | -0.091 | -0.097 | **-0.036** |
| SCAN | 0.652 | **0.774** | 0.631 | 0.038 | 0.088 | **0.031** | -0.022 | -0.085 | **-0.011** |
| Average | 0.672 | **0.789** | 0.664 | 0.076 | 0.088 | **0.037** | -0.022 | -0.062 | **0.006** |
| Networks | ubRMSE (m³/m³) | | | KGE | | | SRC | | |
| | CCI | GLEAM | RF_SMAP | CCI | GLEAM | RF_SMAP | CCI | GLEAM | RF_SMAP |
| AMMA-CATCH | 0.034 | 0.031 | **0.029** | -0.937 | -1.235 | **-0.184** | 0.896 | 0.927 | 0.867 |
| ARM | **0.034** | 0.042 | 0.035 | **0.454** | 0.389 | 0.427 | **0.707** | 0.670 | 0.603 |
| HOAL | 0.025 | 0.024 | **0.023** | 0.635 | 0.547 | **0.687** | 0.622 | **0.769** | 0.746 |
| HOBE | 0.025 | **0.013** | 0.028 | **0.574** | 0.082 | 0.535 | 0.672 | **0.917** | 0.685 |

| | | | | | | | | | |
|---|---|---|---|---|---|---|---|---|---|
| MAQU | 0.041 | **0.038** | 0.041 | 0.555 | 0.323 | **0.570** | 0.556 | **0.663** | 0.536 |
| NGARI | 0.035 | 0.026 | **0.024** | 0.002 | **0.500** | 0.127 | 0.482 | **0.828** | 0.412 |
| ORACLE | 0.071 | 0.057 | 0.071 | 0.233 | **0.392** | 0.275 | 0.562 | **0.880** | 0.574 |
| PBO_H2O | 0.025 | 0.016 | **0.016** | 0.600 | 0.378 | **0.617** | 0.866 | **0.923** | 0.790 |
| REMEDHUS | 0.034 | 0.041 | **0.031** | 0.075 | -0.318 | **0.748** | 0.743 | **0.795** | 0.763 |
| SASMAS | 0.052 | **0.037** | 0.053 | 0.169 | 0.298 | **0.402** | 0.748 | **0.849** | 0.684 |
| SCAN | 0.031 | **0.026** | 0.032 | 0.449 | **0.458** | 0.348 | 0.686 | **0.781** | 0.603 |
| Average | 0.037 | **0.032** | 0.035 | 0.255 | 0.165 | **0.414** | 0.685 | **0.818** | 0.660 |

To evaluate the accuracy of the predicted RF_SMAP dataset in different continents, the average accuracy in each continent was calculated based on the evaluated results in Experiment 2. It should be illustrated that the accuracy at the PBO_H2O was not involved in the calculation due to the scattered distribution of the stations in different continents. As shown in Figure 11, the RF_SMAP dataset can provide the smallest RMSE in each continent among the three datasets. The ubRMSE of the RF_SMAP dataset is smaller than that of the CCI dataset in Africa (AF), Europe (EU), and Asia (AS), revealing that the overall error of the RF_SMAP dataset is smaller than that of the CCI dataset in these continents. In North America (NA) and Oceania (OC, mainland Australia), the accuracy of the CCI dataset is superior to that of the RF_SMAP dataset according to the CC and ubRMSE. Meanwhile, it is seen from the CC and SRC that the GLEAM dataset can more accurately capture the dynamic changes of SM in each continent than the CCI and RF_SMAP datasets. According to the KGE, however, the RF_SMAP dataset is more satisfactory than the CCI and GLEAM datasets in AF, EU, and OC.

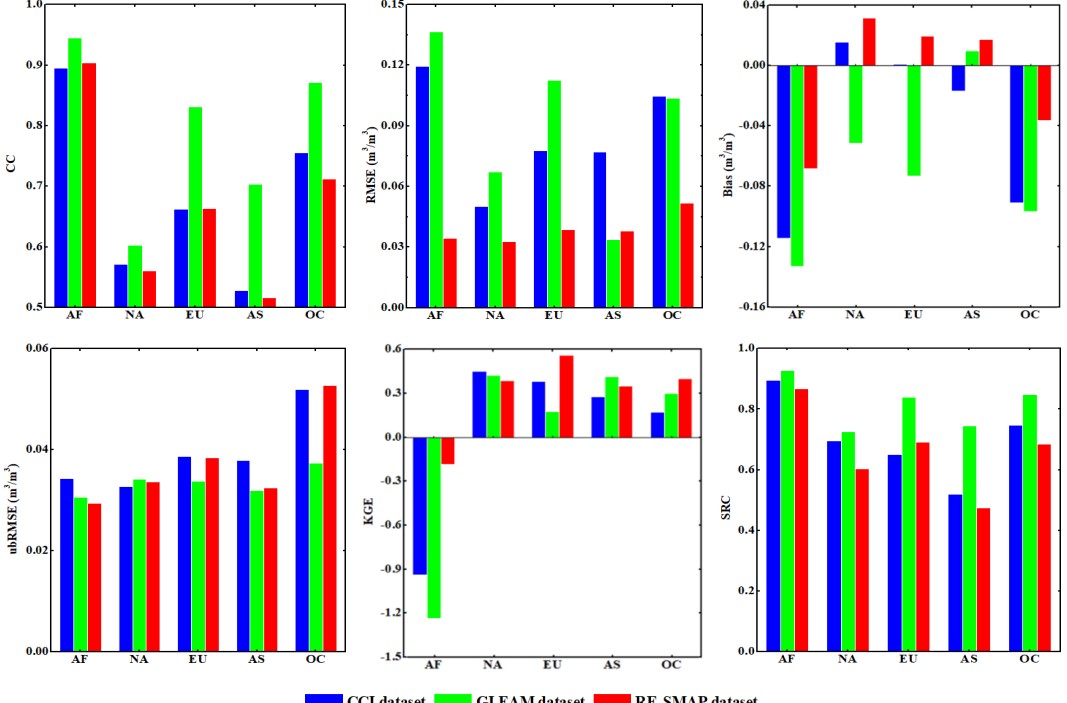

**Figure 11. Accuracy comparison of the historical CCI, GLEAM, and RF_SMAP data in different continents.**



## 4. Discussion

### 4.1. The advantage of the RF model in terms of producing seamless SM data

McNally et al. (2016) pointed out that the spatial coverage of the CCI SM data in eastern AF was generally limited prior to 1992, presenting noticeable gaps in the CCI SM images. Although with the development of sensor technologies, the spatial coverage of the CCI dataset has increased gradually, there are still gaps in parts of central AF and northern SA and several other regions. Moreover, the gaps in some high latitude areas cannot be avoided in the CCI dataset, as shown in the left column of Figure 12. Hence, the quality of the historical CCI dataset has always been constrained by this problem, which will affect greatly the reliable analysis of SM at the global scale. However, this problem is alleviated remarkably in the RF_SMAP dataset. As shown in Figure 12, the spatial coverage (e.g., the regions in the blue ellipses) in AF, SA and some high latitude areas for the RF_SMAP dataset is generally complete. This can be attributed mainly to the spatially complete coverage of the extracted characteristics. Specifically, based on the RF model, the nonlinear relationship between the data before April 2015 and the training characteristics (i.e., HCs extracted from the CCI time-series, TCs and LCs) is constructed, and is migrated to the matched input of testing characteristics (i.e., HCs extracted from the SMAP time-series, TCs and LCs). As the 8-day composited SMAP dataset is generally seamless, the RF_SMAP data can also be predicted with seamless spatial coverage. This is an important advantage of the proposed RF model.

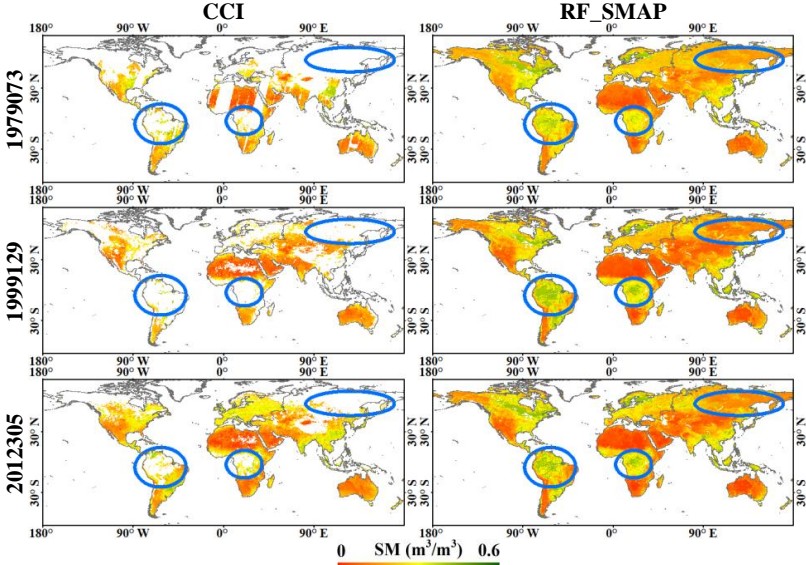

**Figure 12. The spatial coverage of the CCI SM and RF_SMAP dataset (the regions in the blue ellipses indicate the areas with gaps in the CCI dataset but complete spatial coverage in the RF_SMAP dataset).**

### 4.2. Monthly changes in global average SM

To evaluate the monitoring capacity of the predicted RF_SMAP data, the monthly change in global average SM was calculated for the three SM datasets (i.e., SMAP, CCI, and RF_SMAP) in the overlapping period (i.e., 2015105 to 2016097), as shown in Figure 13. It is clearly illustrated that the pattern of monthly changes in the RF_SMAP dataset is more similar to the SMAP dataset than the CCI dataset. Table 7 lists the quantitative evaluation for CCI and RF_SMAP datasets, where the SMAP dataset was used as a reference. The RF_SMAP dataset has a CC of 0.887, which is 0.239 larger than that of the CCI dataset. Furthermore, the RF_SMAP dataset has a RMSE of 0.006 and an ubRMSE of 0.003, which is 0.010 and 0.002 smaller than that of the CCI dataset, respectively. In addition, the Bias of the RF_SMAP dataset is 0.006, which is closer to the reference than that of the CCI dataset (with a Bias value of -0.016).



It should be emphasized that for fairness, the common effective part of the three datasets was used to calculate the global average SM.

Obviously, the average SM increases from May and reaches its peak in July. Then, the average SM begins to decrease and reaches a previous level around September. This phenomenon is caused by seasonal changes in precipitation; accordingly, the pattern of changes in average SM is similar to that of the average precipitation at the global scale (Wood et al., 2015; Konapala et al., 2020; Pascolini-Campbell et al., 2021). Meanwhile, both the SMAP and RF_SMAP datasets describe the seasonal changes in SM. On the whole, although the SMAP SM values are slightly larger than the RF_SMAP SM values, the RF_SMAP dataset can still replace the

SMAP dataset to explain the periodic changes in SM, which can be a more appropriate choice than the CCI dataset.

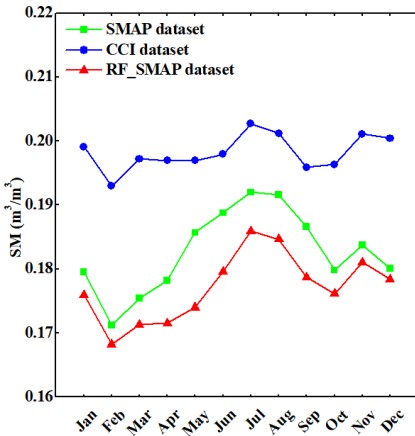

**Figure 13. Monthly changes in the global average SM for the SMAP, CCI, and RF_SMAP datasets (calculated based on data from 2015105 to 2016097).**

**Table 7. Statistical metrics for the monthly changes in SM for the CCI and RF_SMAP datasets (SMAP as reference).**

|  | CCI | RF_SMAP |
|---|---|---|
| CC | 0.648 | **0.887** |
| RMSE ($m^3/m^3$) | 0.016 | **0.006** |
| Bias ($m^3/m^3$) | -0.016 | **0.006** |
| ubRMSE ($m^3/m^3$) | 0.005 | **0.003** |

**4.3. Average SM for different continents**

Each continent has different climatic types and patterns of precipitation. The RF_SMAP dataset can be used to calculate integrally the differences in SM between the continents without the interference of spatial gaps. Therefore, as shown in Figure 14a, we calculated the annual average SM of different continents using the RF_SMAP dataset from 1979 to 2015 to compare each continent. In addition, the average for all 36 years was also provided in Figure 14b. From Figure 14a, we can see that the annual SM for all

continents in the 36 years is generally stable. As shown in Figure 14b, South America (SA) has the largest average SM of 0.261 ($m^3/m^3$) among the six continents. The average SM in North America (NA) and Europe (EU) is similar and slightly smaller than that in South America. The average SM in Asia (AS) is the fourth largest, with a value of 0.188 ($m^3/m^3$), which is 0.035 larger than that in Africa (AF). Oceania (OC, mainland Australia) has the smallest average SM of 0.128 ($m^3/m^3$).



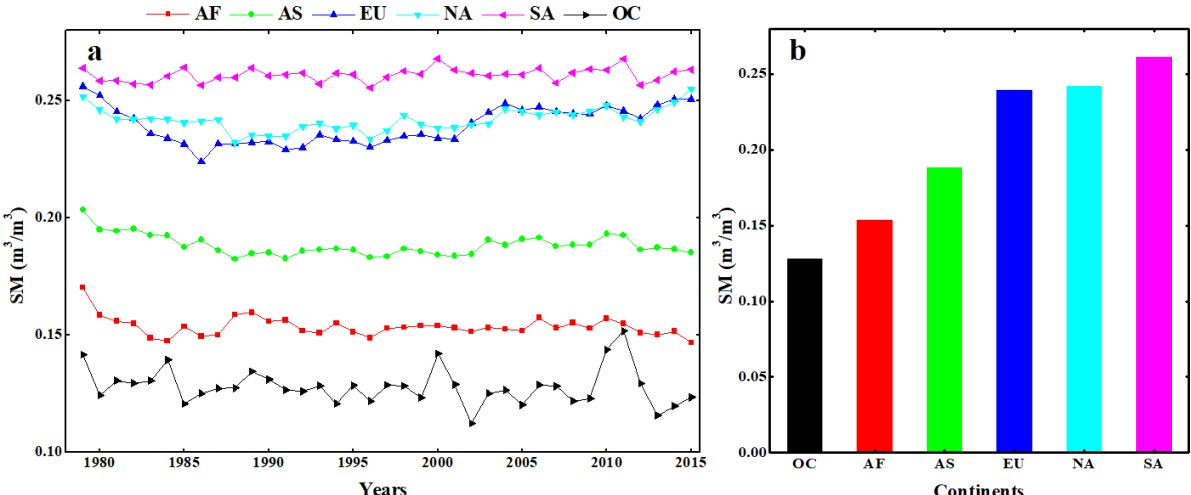

**Figure 14. Average SM (from 1979 to 2015) in the six continents. (a) Annual average SM. (b) Average SM.**

**4.4. Comparison between the predicted RF_SMAP and *SoMo.ml* dataset**

To compare the accuracy between the predicted RF_SMAP and *SoMo.ml* datasets, three networks from January 2000 to December 2004 were used as reference (i.e., ARM, ORACLE, and SCAN). As shown in Table 8, all six indicators of the *SoMo.ml* dataset present greater accuracy than those of the RF_SMAP dataset in the three networks. The reason is that the *SoMo.ml* dataset generated by Sungmin and Orth (2021) has already used *in-situ* data in the model, which can certainly present greater similarity to the *in-situ* data as reference. This also provides a potential solution for enhancing the current RF_SMAP datasets in future research, that is, to calibrate them using the *in-situ* data.

**Table 8. Statistical metrics for accuracy comparison between the RF_SMAP and *SoMo.ml* datasets (the selected *in-situ* data as reference).**

| Network | Dataset | CC | RMSE | Bias | ubRMSE | KGE | SGC |
|---|---|---|---|---|---|---|---|
| ARM | RF_SMAP | 0.828 | 0.052 | 0.048 | 0.019 | 0.358 | 0.847 |
| | *SoMo.ml* | **0.909** | **0.027** | **0.017** | **0.021** | **0.396** | **0.908** |
| ORACLE | RF_SMAP | 0.533 | 0.101 | 0.061 | 0.081 | 0.201 | 0.845 |
| | *SoMo.ml* | **0.922** | **0.058** | **0.008** | **0.058** | **0.439** | **0.901** |
| SCAN | RF_SMAP | 0.620 | 0.032 | 0.006 | 0.032 | 0.373 | 0.602 |
| | *SoMo.ml* | **0.818** | **0.043** | **-0.036** | **0.024** | **0.566** | **0.816** |
| | AVE$_{RF\_SMAP}$ | 0.660 | 0.062 | 0.038 | 0.044 | 0.311 | 0.765 |
| | AVE$_{SoMo.ml}$ | **0.883** | **0.043** | **-0.004** | **0.034** | **0.467** | **0.875** |

**4.5. The uncertainty in the prediction process**

There are three unavoidable uncertainties in the prediction process. First, the RF-based learning model was constructed using the CCI datasets with spatial gaps. The uncertainty in the prediction process is especially large for areas where the CCI data are not available, as the SMAP data were predicted mainly by referring to a relation fitted using CCI data in other areas (e.g., the spatial texture there varies greatly from the gap areas). This issue is prominent when the size of gap is large, where the number of effective training data is also reduced. Second, the RF model is applied based on the assumption that the fitted relationship between the CCI data before April 2015 and the extracted characteristics from CCI data can be migrated to that from the SMAP data. This is supported by the similar pattern of temporal changes of the CCI and SMAP data (as illustrated in Figure 2) as well as the experimental validation. However, it should be pointed out that the relation fitted using the CCI data may not be perfect for SMAP, considering the obvious differences between the two types of data. Although the proposed RF model has been demonstrated to be an effective

solution for creating long time-series SMAP data before April 2015, more efforts are still encouraged to further enhance the accuracy of the predictions in future research. For example, it may be interesting to develop models to construct the relationship between overlapping CCI and SMAP data, but how to fully account for the information in the CCI and SMAP time-series would be an important issue. It would also be important to make fuller use of the available spatial texture information. That is, the spatial content information (e.g., neighborhood information) can be considered in the input construction in the learning model. Third, with the development of computing power, several deep learning-based methods have been developed (Fang et al., 2017; Breen et al., 2020),

which has the potential to fit the complex relationship between various SM datasets. In the future, it would be of great interest to investigate these methods to further increase the accuracy of simulated historical SMAP time-series.

**5. Conclusion**

In this paper, we predicted global 36 km, 8-day composited SM data from 1979 to 2015 based on the development of RF models. We assumed that the CCI dataset has a similar pattern of temporal changes to that of the SMAP dataset. In total, three types of

characteristics were used as the input of the training data in the RF model, including HCs extracted from the CCI time-series (from 2015105 to 2019361), TCs, and LCs. The nonlinear relationships constructed from the characteristics of the CCI dataset were migrated to that of the SMAP dataset. Based on the fitted RF model, the SMAP data between 1979 to 2015 were predicted. Disparate networks of *in-situ* data were used to validate the RF model as well as the predicted RF_SMAP data. The experimental results showed that the average RMSE, Bias and KGE of the RF_SMAP dataset are more satisfactory than those of the widely used

GLEAM and CCI datasets, especially in Africa, Europe, and Oceania. In addition, the predicted RF_SMAP dataset maintained the advantage of the SMAP dataset in terms of spatial accuracy and characterizing pattern of temporal changes. More importantly, the RF_SMAP dataset enlarges the temporal span of current SMAP observations to the same as that of the long time-series CCI SM dataset (i.e., from 1979 to 2015). Furthermore, compared with the CCI SM dataset with many spatial gaps, the predicted RF_SMAP dataset is spatially more complete. Therefore, we conclude that the predicted RF_SMAP dataset is a reliable substitute for the CCI

SM dataset. The RF_SMAP dataset will be available at https://doi.org/10.6084/m9.figshare.17621765 to facilitate free usage of the data.

**Data availability**

The predicted RF_SMAP dataset are available at https://doi.org/10.6084/m9.figshare.17621765 (Yang et al., 2021).

**Author contributions**

HY designed the research, analyzed the data, wrote the original manuscript, and produced the dataset. QW revised the whole manuscript and provided the funding to support the research. WZ and PMA provided direction and comments. All authors edited and approved the final manuscript.

**Competing interests**

The authors declare that they have no conflict of interest.



**Acknowledgment**

The authors like to thank the NSIDC, Global Energy and Water Cycle Experiment (GEWEX), and European Space Agency (ESA) for making the SMAP, CCI and ISMN data freely available.

**Financial support**

This research was supported by the National Natural Science Foundation of China under Grants 42222108, 42171345, 41971297
and 42221002.

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
