# Peer review of "An 8-day composited 36 km SMAP soil moisture dataset from 1979 to 2015 produced using a random forest and historical CCI data"

_Earth System Science Data, 2022_

## Referee Comment (RC1)

This study develops a new global long-term soil moisture dataset by extending the SMAP data back into the ESA CCI era. To do this, the authors train a random forest with historical CCI data and apply the trained model to estimate SMAP soil moisture under the assumption that CCI and SMAP soil moisture have similar temporal variability. Such a dataset is valuable and relevant for a variety of climatological and hydrological applications.

However, the main assumption made in this study requires more careful investigation. The authors present the temporal variability of CCI and SMAP over an overlapping period of 4-5 years, but most of the selected sites appear to be in arid regions (Fig. 2); comparison needs to be made in a more comprehensive way, including humid and/or high latitude regions with relatively high variability of soil moisture. Moreover, how can we ensure that this similarity is preserved also during the previous ~30 years of the CCI era?

Second, the method described in Sect. 2.2.2 and 2.2.3 needs more explanation/clarification here and there. For instance, what is the purpose of having two separate experiments? If Experiment 1 was to evaluate the performance of RF_SMAP during the period of SMAP, the SMAP should be included for the model evaluation, e.g. in Fig. 6, Fig. 7, Table 5.

Why is original soil moisture data (I guess you mean actual soil moisture time-series) unsuitable for model training? The authors stated that this is because 1) SM data has spatial gaps and 2) abundant precipitation can lead to abnormal change in SM (Lines 147-149). However, RF can only be trained with grid pixels where SM data is available, and the diverse relationship between precipitation-soil moisture should be included in the training data.

To predict RF_SMAP, the trained RF model uses characteristics extracted from SMAP as input (Lines 185-194) at each grid pixel, is this correct? Then, how did you generate RF_SMAP for pixels and periods that do not have SMAP data (and thus unable to extract characteristics from SMAP)?

Lastly, the validation of RF_SMAP over the CCI era is highly limited due to the lack of ISMN before 2000. The validation of RF_SMAP over diverse climate regimes also seems limited, as most ISMN data are obtained from the US. I also wonder if there are any systematic biases between the RF_SMAP (historical SMAP before 2015) and the actual SMAP data from 2015.

It is not clear why e.g. Fig. 6 shows only one time series per network and Fig. 7 shows a very small number of samples (dots) given that each ISMN network has >400 stations according to Table 2. Moreover, the comparison between the gridded datasets could be done from more diverse perspectives, e.g. comparison by season, during extreme (drought) conditions; SoMo is global, long-term data, but the comparison is done only for 4 years at three locations (Sect. 4.4)

Given that these aspects require substantial consideration, I think that the current manuscript is not publishable in ESSD.

---

## Community Comment (CC2)

This study develops a new global long-term soil moisture dataset by extending the SMAP data back into the ESA CCI era. To do this, the authors train a random forest with historical CCI data and apply the trained model to estimate SMAP soil moisture under the assumption that CCI and SMAP soil moisture have similar temporal variability. Such a dataset is valuable and relevant for a variety of climatological and hydrological applications.

*Dear Referee*

*Thank you for your comments. The comments are undoubtedly helpful to improve the quality of the paper. Accordingly, we have analyzed the comments carefully and provided the response below.*

However, the main assumption made in this study requires more careful investigation. The authors present the temporal variability of CCI and SMAP over an overlapping period of 4-5 years, but most of the selected sites appear to be in arid regions (Fig. 2); comparison needs to be made in a more comprehensive way, including humid and/or high latitude regions with relatively high variability of soil moisture. Moreover, how can we ensure that this similarity is preserved also during the previous ~30 years of the CCI era?

*Reply:*

*In fact, these five pixels in Figure2 were randomly selected. **We consider that this issue can be solved by providing a more complete description in Section 2.2.3. Specifically, based on different humid regions and latitudes, we are going to supply more pixels to exhibit the change pattern of SM in Figure 2.***

*As for the preservation of similarity during the previous ~30 years, we can explain this point from three aspects. First, the similarity of the CCI and SMAP data from 2015 to 2019 has been exhibited in Figure 2, which shows great similarity already. Second, we have found that both the CCI and predicted SMAP data can preserve consistent similarity to the in-situ data from a large period of about 10 years (i.e., from 1996 to 2015, as the earliest in-situ data began in 1996). Third, for the period before 1996, although there were no in-situ and SMAP data available for comparison with CCI data, the experimental results indicated that the general temporal profiles of CCI and predicted RF_SMAP are similar. Thus, we believe that the similarity can also be preserved over the 30 years.*

***To support the main assumption (similarity of the CCI and SMAP datasets), Figure 2 was modified and provided here in advance. Based on the original 6 pixels, we supplied additional 12 pixels (16 pixels in total). The pixels have random distribution, which include arid regions (e.g., Pixel 1 and 9), high latitudes (e.g., Pixel 10), and high altitudes (e.g., Pixel 12).***

[Figure]

[Figure]

*Figure 2. The patterns of changes in SMAP and CCI SM in temporal and spatial domains (16 pixels selected randomly at the global scale)*

Second, the method described in Sect. 2.2.2 and 2.2.3 needs more explanation/clarification here and there. For instance, what is the purpose of having two separate experiments? If Experiment 1 was to evaluate the performance of RF_SMAP during the period of SMAP, the SMAP should be included for the model evaluation, e.g. in Fig. 6, Fig. 7, Table 5.

*Reply:*

*As for this issue, **we are going to supply more detailed description in Section 2.2.3**. **Meanwhile, the main purpose of the two experiments will also be further clarified in Section 2.2.3.** In fact, Experiment 1 aimed to demonstrate the predicted method based on the in-situ as well as the real SMAP data. Specifically, by using the CCI and SMAP data from 2016 to 2019 (2016105 to 2019361), the predicted data (i.e., RF_SMAP) in 2015 (2015105 to 2016097) were generated. In Experiment 1, both the real SMAP and in-situ data in 2015 (2015105 to 2016097) were available for validation. Accordingly, Figure 6, Figure 7 and Table 5 were the model evaluation results by referring to the in-situ data. Figure 5 and Table 4 are the evaluation results that used the SMAP data as reference. The reason of designing the experiment using the real SMAP as reference is that the reference in this case is known perfectly, avoiding the uncertainty introduced by other factors (e.g., the uncertainty in spatial support and geographical location in in-situ data).*

*As for Experiment 2, it aimed to validated the RF_SMAP dataset from 1979 to 2015 based on the in-situ data, as in this period only the real SMAP data are not available.*

Why is original soil moisture data (I guess you mean actual soil moisture time-series) unsuitable for model training? The authors stated that this is because 1) SM data has spatial gaps and 2) abundant precipitation can lead to abnormal change in SM (Lines 147-149). However, RF can only be trained with grid pixels where SM data is available, and the diverse relationship between precipitation-soil moisture should be included in the training data.

*Reply:*

*It should be illustrated that the original SMAP time-series were used for model training in the first version (ESSD_2022_137). However, in the process of revision, we found the shortcomings of this training method. Specifically, in high latitude regions, the original SMAP time-series data contain unavoidable gaps (i.e., the missing data) in a year because of the snow cover and other factors. Theoretically, these spatially missing data cannot be involved in the training process, as you mentioned exactly. If we want to directly use the SMAP time-series data for training, we need to mask the regions with gaps. However, the usage of the mask can significantly harm the reliability of the RF_SMAP dataset in terms of spatial coverage. Also, the number of training data in the RF model can be reduced greatly. Hence, the hctsa characteristics-based training method was adopted in the manuscript. Since the hctsa-extracted temporal characteristics are spatial seamless, the interference of missing data in the SMAP time-series on model training can be eliminated.*

To predict RF_SMAP, the trained RF model uses characteristics extracted from SMAP as input (Lines 185-194) at each grid pixel, is this correct? Then, how did you generate RF_SMAP for pixels and periods that do not have SMAP data (and thus unable to extract characteristics from SMAP)?

*Reply:*

*We need to clarify that the trained RF model did not use the characteristics extracted from SMAP as input.*

*In fact, the construction of model is based on the core assumption that the CCI and SMAP datasets have similar pattern of temporal changes. Specifically, the model at a time t was trained by the label ($CCI_t$) and the characteristics (extracted **from the CCI time series by the hctsa method**, coupled with the DEM and location data). In the prediction process, the characteristics (extracted **from the SMAP time series by the hctsa method**, coupled with the DEM and location data) were imported into the trained model, and the $SMAP_t$ data at time t was predicted. With the continuous change of $CCI_t$ data from 1979001 to 2015097 (i.e., t, t+1, t+2, t+3, ...), different RF models were continuously trained and corresponding $RF\_SMAP_t$ data were predicted in turn. **We are going to rewrite this point and add key information in the new version of Figure 3.***

*To clearly illustrate the prediction process, Figure 3 is modified in advance.*

[Figure]

*Figure 3. The prediction process of the RF_SMAP dataset at a time.*

Lastly, the validation of RF_SMAP over the CCI era is highly limited due to the lack of ISMN before 2000.The validation of RF_SMAP over diverse climate regimes also seems limited, as most ISMN data are obtained from the US. I also wonder if there are any systematic biases between the RF_SMAP (historical SMAP before 2015) and the actual SMAP data from 2015.

*Reply:*

*As you mentioned exactly, due to the uneven distribution and lack of in-situ stations, it is difficult validate the dataset based on diverse climate regions. However, the comparison based on different periods (e.g., from 2000 to 2005, and 2010 to 2015) is possible, **we are going to analyze the systematic biases between the RF_SMAP (historical SMAP before 2015) and the actual SMAP data from 2015. We agree that this point is valuable, and we will revise.***

It is not clear why e.g. Fig. 6 shows only one time series per network and Fig. 7 shows a very small number of samples (dots) given that each ISMN network has >400 stations according to Table 2. Moreover, the comparison between the gridded datasets could be done from more diverse perspectives, e.g. comparison by season, during extreme (drought) conditions; SoMo is global, long-term data, but the comparison is done only for 4 years at three locations (Sect. 4.4)

*Reply:*

*First of all, we need to clarify that Figure 6 aimed to show the change pattern of 11 networks (11 sub-figures) at 46 prediction times (i.e., from 2015105 to 2016097). Figure 7 provided the scatter plot of the corresponding 11 networks at 46 prediction times, based on the results in Figure 6. We need to clarify that the validation was at network level, that is, all stations in a network were averaged. In fact, the number of samples in Figure 7 is 46 (i.e., the prediction times) rather than the number of stations. **The authors are going to revise the corresponding description to clarify this confusion.***

*Additionally, the comparison of datasets in terms of different seasons is interesting. We will provide the results accordingly in the new version.*

*As for the comparison with the SoMo.ml dataset in Section 4.4, we need to clarify the purpose of this section first. That is, Section 4.4 aimed to exhibit the differences between the SoMo.ml and RF_SMAP dataset and provide a potential way to improve the RF_SMAP dataset in future. Specifically, the production of the SoMo.ml dataset used the in-situ data as model inputs to improve the accuracy. However, the in-situ data are always used as the reference for validation, which is undoubtedly beneficial for accuracy evaluation of the SoMo.ml dataset. In Section 4.4, we admitted the difference in accuracy between the SoMo.ml and RF_SMAP dataset, and proposed to use in-situ data to further enhance the predicted RF_SMAP dataset in future research. Thus, we considered that using longer time series of the SoMo.ml data and more in-situ data will not add anything to the current points in Section 4.4 (i.e., the conclusions will also be the same as the current version).*

---

## Community Comment (CC3)

This study provides an accurate and reliable, global 36 km, 8-day synthetic SMAP SM products from 1979 to 2015. This is a valuable dataset for the evaluation of historical events. I have some questions about the dataset you achieved in your article

Thank you for the comments. We are going to reply item-by-item to each comment.

1.How to achieve data synthesis toward those different data sources?

*Reply:*

*Considering the spatial gaps in the daily SMAP data, we adopted 8-day composited method to acquire a more complete spatial coverage by averaging the valid SM data. Thank you for the comment.*

2.It seems that the analysis of data in this volume is time-consuming. Could you please provide more details about the data analysis platform that you used here? Such as software or any other online platform.

*Reply:*

*Indeed, the time cost of this work is related high. We used Matlab to generate the RF_SMAP dataset. For the simulation of one scene data, it takes about 300 seconds. The processed operator is Intel(R) Xeon(R) Silver 4110 CPU @ 2.10GHz.*

In addition, there are some minor issues with the manuscript details:

1.In Figure 1, the site location is not clear.

*Reply:*

*Thank you for the comment, we have revised the Figure 1 and enlarged the size of the sample points to clear illustrate the site location.*

[Figure]

Figure 1. Locations of the in-situ data in Experiments 1 and 2.

2.Is the reconstruction of SM data before 2015? Why do you use sites from 2015-2016 to validate pre-2015 data in the abstract?

*Reply:*

*Thank you for the suggestion. For the unclear description in the abstract, we are going to revise revised. In fact, the validation of reconstruction before 2015 was adopted the in-situ data before 2015 as the reference (Experiment 2). The in-situ data from 2015-2016 were used in Experiment 1. Experiment 1 aimed to demonstrate the predicted method (evaluate the performance of RF_SMAP during the period of real SMAP).*

3.I think the flowchart is kind of too simple to express the details of dataset production.
*Reply:*

*Thank you for the suggestion. We have revised this point in advance and increased the readability of Figure 3.*

[Figure]

*Figure 3. The prediction process of the RF_SMAP dataset at a time.*

---

## Author Comment (AC3)

The manuscript presents a composited, SMAP-like soil moisture dataset derived with a random forest approach from historical CCI data. While a global soil moisture product for such a long period (1979 to 2015) has its merits, the dataset stands and falls with the successful evaluation of the derived product. Its spatial resolution (36 km) is interesting for large scale analyses.

It should be noted that the approach a) assumes stationarity of the CCI and SMAP data and b) generalises its global applicability of the model. Both are rather strong assumptions for a "simple" random forest approach. In addition, the evaluation refers to averaged network-level data, which introduces further uncertainty of the scaling. The authors present some limited evaluation, which does not really exemplify the validity of the derived model data.

There are some similar studies and datasets, the authors did not refer to: https://www.nature.com/articles/s41597-023-02053-x , https://www.nature.com/articles/s41597-021-00925-8 In which way does their dataset (and approach) advance these?

Moreover, there is a recent paper in HESS, which uses Sentinel data to estimate soil moisture: https://hess.copernicus.org/articles/27/1221/2023/

Obviously, the temporal extent of this approach has very little overlap with the data presented in this manuscript (since the sentinel satellites have been operational just from 2015 onwards). However, the authors might find inspiration for further evaluation in this?

After all, it is very difficult to evaluate if the dataset and its presentation justify publication in ESSD. If the authors could really corroborate the validity of their data product, this would be a clear yes. However given the open questions and despite the meticulous effort which went into the compilation of this data, it remains too unclear, how the dataset from a rather simple approach can advance already existing SMAP-like soil moisture products.

*Reply:*
*Dear Referee*
*   Thank you very much for your comments. The comments are undoubtedly helpful to improve the quality of the paper. Accordingly, we have analyzed the comments carefully and provided the response below.*
*   The major comments include three aspects. **First, the referee concerns the stationarity of the CCI and SMAP datasets. Second, the referee concerns the global applicability for a simple RF model. Third, the referee concerns that an averaged network-level validation can introduce further uncertainty.** To address the above questions, we have provided responses and preliminary modifications, respectively.*
*   First, both the CCI and SMAP datasets are used to characterize the surface soil moisture, which have the same units (i.e., $m^3/m^3$) and physical meaning. Meanwhile, the original data of the CCI and SMAP datasets are derived from satellite sensors. Thus, it is reasonable to assume that the CCI and SMAP datasets have similar change pattern in a long-time series. Additionally, we have provided the basis for the stability of the CCI and SMAP datasets in the original manuscript*

*(Figure 2).* **For this concern, we are going to add more pixels in different regions to clearly illustrate the stability of the CCI and SMAP datasets.**

*To clearly illustrate this point, we revise **Figure 2** and supply more pixels to exhibit the stationarity of the CCI and SMAP datasets in advance.*

[Figure]

*Figure 2. The patterns of changes in SMAP and CCI SM in temporal and spatial domains (16 pixels selected randomly at the global scale)*

*Second, it should be clarified that the proposed RF model is dynamic, and needs to continuously train for different scenes in a long-time series. Moreover, for high latitudes and high altitudes, spatial gaps usually affect the data quality. To solve this issue, the hctsa-based method was adopted in the manuscript to extract spatially seamless characteristics, suggesting that the spatially seamless RF_SMAP dataset can be generated.*

*The construction of model is based on the core assumption that the CCI and SMAP datasets have similar patterns of temporal changes. Specifically, the model at a time t was trained by the label ($CCI_t$) and the characteristics (extracted from the CCI time series by the hctsa method, coupled with the DEM and location data). In the prediction process, the characteristics (extracted*

*from the SMAP time series by the hctsa method, coupled with the DEM and location data) were imported into the trained model, and the SMAP$_t$ data at time t was predicted. With the continuous change of CCI$_t$ data from 1979001 to 2015097 (i.e., t, t+1, t+2, t+3, ...), different RF models were continuously trained and corresponding RF_SMAP$_t$ data were predicted in turn.* **We are going to rewrite this point in the updated manuscript and provide more detailed information in the new version of Figure 3.**

**To clearly illustrate the prediction process, Figure 3 is modified in advance.**

[Figure]

*Figure 3. The prediction process of the RF_SMAP dataset at a time.*

*Currently, the training method is developed using the globe-based data.* **Considering the concern of the referee, we will add an additional training method** *(i.e., continent-based data are used for training to generate datasets in different continents)* **for comparison in Section 3.3.**

**The comparison using different training data in Section 3.3 is provided in advance.** *Specifically, 46 scenes in each continent from 2015105 to 2016097 were predicted using the different training data (including the globe- and continent-based data). By referring to the true SMAP data in each continent, it can be seen from Table 7 that the predicted results using the globe-based data (used in the manuscript) is more accurate than those using the continent-based data. More precisely, using the globe-based data in AF can provide the CC of 0.941 and the RMSE of 0.041, which are more satisfactory than using the continent-based data. Moreover, the CC values using the globe-based data in EU and OC are 0.871 and 0.881, which are 0.002 and 0.018 higher than those using the continent-based data. In all, using the globe-based data is more effective than using the continent-based data. This means that the spatially adjacent information in the training data is useful to improve the prediction accuracy of the dataset.*

*Table 7. Comparison between the use of different train data.*

| Continent | Training data | CC | RMSE | Bias | ubRMSE |
|---|---|---|---|---|---|
| AF | Globe-based | **0.941** | **0.041** | **0.001** | **0.041** |
| | Continent-based | 0.935 | 0.043 | 0.001 | 0.042 |
| AS | Globe-based | **0.921** | **0.058** | **0.010** | **0.057** |
| | Continent-based | 0.920 | 0.059 | 0.010 | 0.057 |
| EU | Globe-based | **0.871** | **0.061** | **0.006** | **0.058** |

| | | | | | |
|---|---|---|---|---|---|
| | *Continent-based* | *0.869* | *0.063* | *0.007* | *0.059* |
| *NA* | ***Globe-based*** | ***0.913*** | ***0.064*** | ***0.015*** | ***0.061*** |
| | *Continent-based* | *0.911* | *0.065* | *0.016* | *0.063* |
| *OC* | ***Globe-based*** | ***0.881*** | ***0.048*** | ***-0.001*** | ***0.047*** |
| | *Continent-based* | *0.863* | *0.052* | *-0.011* | *0.051* |
| *SA* | ***Globe-based*** | ***0.910*** | ***0.060*** | ***0.010*** | ***0.059*** |
| | *Continent-based* | *0.909* | *0.061* | *0.010* | *0.059* |

*Third, **we are going to supply a station-level evaluation for different climate types in Experiment 2 to further demonstrate the advantage of the RF_SMAP dataset.** Meanwhile, the supplement can clearly illustrate the validity of the RF_SMAP dataset and can provide a basis for applications. In addition, the uncertainty of validation can also be further reduced.*

*    **To further validate the RF_SMAP dataset, we provide the station-level evaluation in Figure 12 based on the Köppen-Geiger climate classification in advance.** Generally, we found that the station-level evaluated results are similar to the network-level evaluated results. According to the KGE, RMSE, and ubRMSE, the RF_SMAP dataset can provide a more satisfactory performance than the CCI and GLEAM datasets in the arid steppe and temperate regions. The KGE (RMSE) of the GLEAM dataset is always smaller (higher) than that of the CCI and RF_SMAP datasets in the cold regions. Furthermore, the Bias of the RF_SMAP dataset is usually closer to the reference than that of the CCI and GLEAM datasets in addition to in BSh, Cfa, and Csa regions. Besides, the CC and SRC of the GLEAM dataset are always higher than those of the CCI and RF_SMAP datasets.*

[Figure]

*Figure 12. Accuracy comparison of the historical CCI, GLEAM, and RF_SMAP datasets in different climate types (each climate type contains the different number of stations, BSk contains 118 stations, BSh contains 7 stations, BWh contains 20 stations, BWk contains 39 stations, Cfa contains 83 stations, Csa contains 16 stations, Csb contains 11 stations, Dfa contains 35 stations, Dfb contains 18 stations, Dsb contains 7 stations, Dwc contains 8 stations, respectively. Climate types with less than 5 stations are not counted).*

*    After revising, the accuracy validation of the RF_SMAP dataset is relatively comprehensive, including the network- and station-level. Meanwhile, six statistical metrics (i.e., CC, RMSE, Bias,*

*ubRMSE, KGE, and SRC) has been adopted in the manuscript, two widely used SM datasets (i.e., CCI, and GLEAM) have been used as benchmark, and the validation for different networks, continents and climate types has been provided to support the RF_SMAP dataset.*

*In addition, the three mentioned papers (Yao et al., 2021; Madelon et al., 2023; Skulovich and Gentine, 2023) will be added to the introduction, which can provide important theoretical support. According to the validated strategy in Skulovich and Gentine. (2023), we have added a new station-level validation for different climate types (i.e., Figure 12). Besides, the time period of the RF_SMAP dataset is longer than that in these papers.*

*Madelon, R., Rodríguez-Fernández, N.J., Bazzi, H., Baghdadi, N., Albergel, C., Dorigo, W., & Zribi, M. (2023). Soil moisture estimates at 1 km resolution making a synergistic use of sentinel data. Hydrology and Earth System Sciences, 27, 1221-1242.*

*Skulovich, O., & Gentine, P. (2023). A long-term consistent artificial intelligence and remote sensing-based soil moisture dataset. Sci Data, 10, 154.*

*Yao, P., Lu, H., Shi, J., Zhao, T., Yang, K., Cosh, M.H., Gianotti, D.J.S., & Entekhabi, D. (2021). A long term global daily soil moisture dataset derived from amsr-e and amsr2 (2002-2019). Sci Data, 8, 143.*